# Comparison of Six Serological Immunoassays for the Detection of SARS-CoV-2 Neutralizing Antibody Levels in the Vaccinated Population

**DOI:** 10.3390/v14050946

**Published:** 2022-04-30

**Authors:** Hee-Jung Lee, Jin Jung, Ji Hyun Lee, Dong-Gun Lee, Young Bong Kim, Eun-Jee Oh

**Affiliations:** 1Department of Biomedical Science and Engineering, Konkuk University, Seoul 05029, Korea; ziniga@konkuk.ac.kr; 2Department of Laboratory Medicine, Seoul St. Mary’s Hospital, College of Medicine, The Catholic University of Korea, Seoul 06591, Korea; bluejin1227@gmail.com; 3Research and Development Institute for In Vitro Diagnostic Medical Devices of Catholic University of Korea, Seoul 06591, Korea; 4Department of Biomedicine & Health Sciences, Graduate School, The Catholic University of Korea, Seoul 06591, Korea; onion1002@naver.com; 5Division of Infectious Diseases, Department of Internal Medicine, Seoul St. Mary’s Hospital, College of Medicine, The Catholic University of Korea, Seoul 06591, Korea; symonlee@catholic.ac.kr

**Keywords:** SARS-CoV-2, neutralizing antibody, ACE2-RBD inhibition assay, immunoassay

## Abstract

Neutralizing antibody (NAb) detection is critical for evaluating herd immunity and monitoring the efficacy of vaccines against severe acute respiratory syndrome coronavirus 2 (SARS-CoV-2). In this study, quantitative SARS-CoV-2 antibody levels after vaccination were measured by chemiluminescent immunoassays, enzyme immunoassays, and surrogate virus neutralization tests (sVNTs), as well as plaque reduction neutralization tests (PRNT). Sequential blood samples were collected before and 1 and 3 months after vaccination in 30 healthy participants (two doses of Oxford-AstraZeneca [AZ] or Pfizer-BioNTech [BNT]). After vaccination, all sera tested positive for PRNT, with NAb titers ranging from 1:10 to 1:723. Median NAb titers were higher in the BNT vaccine group than in the AZ vaccine group at both one and three months post-vaccination. Excellent overall concordance rates were observed between serological assays and PRNT. In a quantitative correlation analysis, the results of sVNTs showed a strong correlation with those of PRNT. Results of the four binding antibody assays showed a significant correlation with those of PRNT. The serologic assays evaluated in this study could be used as sVNTs to evaluate the efficacy of SARS-CoV-2 vaccines.

## 1. Introduction

Coronavirus disease 2019 (COVID-19), caused by severe acute respiratory syndrome coronavirus 2 (SARS-CoV-2), has emerged as the most serious global health crisis since the influenza pandemic of 1918 [1]. To date, more than 6.1 million deaths have been reported globally. COVID-19 vaccines based on two different platforms have been granted emergency use authorization by the U.S. Food and Drug Administration (FDA) (Pfizer/BioNtech and Moderna mRNA vaccines and the Janssen viral vector vaccine) and by the UK government (AstraZeneca viral vector vaccine) [2,3]. An optimal COVID-19 vaccine provides a long-lasting antibody response, stimulates immunity, and can prevent disease and forward transmission [4]. These vaccines effectively prevent moderate-to-severe COVID-19 [5].

The detection of specific antibodies, such as receptor-binding domain (RBD)- or spike protein (S1)-specific immunoglobulin (Ig)G and neutralizing antibodies (NAbs) following SARS-CoV-2 infection, plays a key role in evaluating seroprevalence, monitoring herd immunity, and designing vaccination strategies [6,7]. NAbs against SARS-CoV-2 are important because they inhibit the binding of the RBD of the spike protein to human angiotensin-converting enzyme 2 (ACE2) receptor [8,9]. 

The current gold standard for the detection of neutralizing antibodies against SARS-CoV-2 is a virus neutralization test, such as the plaque reduction neutralization test (PRNT). However, PRNT is difficult to perform in routine laboratories because it requires live pathogens and biosafety level (BSL) 3 laboratories. PRNT also has several disadvantages, such as the requirement for technical expertise, time-consuming, expensive, and non-standard protocols, and difficulty for high-throughput analyses [10,11]. Therefore, it is necessary to develop a cost-effective, rapid, and comprehensive alternative NAb detection method. Several serological assays, such as the surrogate virus neutralization test (sVNT) for detecting human ACE 2 receptor-blocking antibody, and S1/RBD-specific binding antibody tests, have been developed to test humoral immunity following SARS-CoV-2 infection or vaccination. Many commercial serological assays based on the principles of enzyme immunoassays (EIA), fluorescent immunoassays (FIA), and chemiluminescent immunoassays (CLIA) have been developed to improve large-scale immunity testing. However, few studies have compared assays to identify effective surrogates for virus neutralization. In the current study, two sVNTs detecting ACE inhibition, two EIAs, and two CLIAs measuring S1/RBD-specific binding antibody levels were compared with PRNT using serum samples from vaccinated populations.

## 2. Materials and Methods

### 2.1. Clinical Samples

This study included 30 participants with no history of infection with SARS-CoV-2 who received two doses of the AstraZeneca ChAdOx1 vaccine (with an 11-week interval between dose 1 and 2) (*n* = 15) or Pfizer BNT162b2 vaccine (3-week interval between dose 1 and 2) (*n* = 15) between March 2021 and June 2021 [12,13]. Participants who developed a SARS-CoV-2 infection during the study, or for whom all three blood samples were not obtained, were excluded from this study. The characteristics of the 30 participants are presented in Table A1. Blood samples were collected from all participants before the first dose and 1 month and 3 months after the second dose of the vaccine. Venous blood samples were collected in 8 mL serum-separating tubes and centrifuged at 2000× *g* for 10 min. All serum samples were stored at 4 °C for up to 2 weeks and aliquoted for assessment. Serum aliquots were stored at −80 °C until use for tests.

Antibodies against SARS-CoV-2 were measured using six serological assays and the PRNT. This study is part of an ongoing single-center study at Seoul St. Mary’s Hospital to determine the immunogenicity of different COVID-19 vaccines in healthcare workers. The study was approved by the Institutional Review Board of Seoul St. Mary’s Hospital (KC21TISI0114). All participants provided written informed consent before participating in the study.

### 2.2. Plaque Reduction Neutralization Test 

VeroE6 cells (an African green monkey kidney cell line) were purchased from the Korean Cell Line Bank (Seoul, Korea). The cells were maintained in Dulbecco’s modified Eagle’s medium (DMEM; Thermo Fisher Scientific, Waltham, MA, USA) containing 10% fetal bovine serum (FBS; Thermo Fisher Scientific) and 1% penicillin–streptomycin (P/S; Thermo Fisher Scientific) and maintained at 37 °C under 5% CO_2_. SARS-CoV-2 (BetaCoV/Korea/KCDC03/2020) was provided by the Korea Disease Control and Prevention Agency (KDCA; Osong, Korea) and was prepared by propagation in Vero E6 cells. Cell culture procedures were carried out in BSL2 and moved to a BSL3 facility for viral infection and assays. 

The diluted serum was mixed with the same volume of virus suspension (100 plaque-forming units, pfu) [14]. After incubation at 37 °C for 1 h, the mixture was added to a monolayer of Vero E6 cells. Adsorption was performed at 37 °C for 1 h, and the cells were then covered with DMEM containing 1.5% SeaPlaque agarose (Lonza, Walkersville, MD, USA), 5% FBS, and 1% P/S. After incubation at 37 °C for 3 days, the plaques were stained with 2% crystal violet solution (Georgia Chemicals, Norcross, GA, USA) and counted. The neutralization titer was calculated as the reciprocal of the highest serum dilution that reduced plaque formation by 50% (PRNT_50_) [15].

### 2.3. SARS-CoV-2 Surrogate Virus Neutralization Test

Two SARS-CoV-2 sVNT kits, the SARS-CoV-2 sVNT kit (GenScript, Piscataway, NJ, USA) and the GenBody FIA COVID-19 NAb kit (GenBody, Chungcheongnam-do, Korea) were used to detect NAbs targeting the RBD or spike protein. GenScript and GenBody sVNT assays measure the ability to inhibit RBD-ACE2 binding based on antibody-mediated blockage of the interaction between the ACE2 receptor and SARS-CoV-2 RBD or spike protein using EIA and FIA, respectively. Cut-offs of ≥30% (GenScript) and ≥25% inhibition (GenBody) were applied according to the manufacturers’ instructions. For quality control, positive and negative control samples were included in each batch. For samples with qualitative discrepancies among assays, repeated tests were performed to confirm the results. Detailed descriptions of the assay kits are presented in Table A2(A).

### 2.4. SARS-CoV-2 Binding Antibody Tests

All samples were tested using the following four SARS-CoV-2 binding antibody assays: Elecsys Anti-SARS-CoV-2 S (Roche Diagnostics, Basel, Switzerland), SARS-CoV-2 IgG (sCOVG) (Siemens, Munich, Germany), AdvanSure SARS-CoV-2 IgG (RBD) ELISA (LG Chem, Seoul, Korea), and AdvanSure SARS-CoV-2 IgG (S1) ELISA (LG Chem, Seoul, Korea). The former two are CLIAs, and the latter two are ELISAs. Detailed descriptions of the assay kits are presented in Table A2(B). All assays were performed according to the manufacturers’ instructions. For the Roche and Siemens assay kits, the samples were retested after an additional dilution step if the values obtained exceeded the analytical measurement interval. According to the WHO international standard for anti-SARS-CoV-2 immunoglobulin, traceable units of bound antibody per milliliter (BAU/mL) were calculated using conversion factors (Roche 1.208: Siemens 21.803). The binding antibody levels obtained by serological PRNT and sVNT were compared using serial serum samples from vaccine recipients.

### 2.5. Statistical Analyses

Continuous data, given as medians (interquartile range [IQR]), were compared using rank sign tests (Mann–Whitney U and Wilcoxon tests). Intra-individual changes in antibody levels were analyzed using paired *t*-tests. Spearman’s rank correlation was used to compare quantitative values obtained by different assays. The agreement between assays was calculated using Cohen’s kappa. Spearman correlation coefficients were calculated for correlations between the SARS-CoV-2 serologic assays, and correlations were defined as strong (0.7–1.0), moderate (0.5–0.7), and weak (0.3–0.5). Statistical analyses were performed using MedCalc 20.006 (MedCalc, Ostend, Belgium), and graphical representations of the data were generated using GraphPad Prism version 9.2.0 for Windows (GraphPad Software, San Diego, CA, USA). Statistical significance was set at *p* < 0.05.

## 3. Results

### 3.1. Neutralizing Antibody Titers by PRNT and sVNTs

All serum samples (*n* = 30) collected before vaccination were negative for PRNT (less than 1:10). All serum samples (*n* = 60) collected after the two-dose vaccination were positive, with NAb titers ranging from 1:10 to 1:723. One month after vaccination, 13.3% (2/15) of the AZ group and 58.8% (10/15) of the BNT group had elevated NAb titers (≥1:60). Three months after vaccination, 6.7% (1/15) of the AZ group and 41.2% (7/15) of the BNT vaccine group still showed high NAb titers (≥1:60). NAb titers ≥ 1:320 were detected only in the BNT group [33.3% (5/15) at 1 month and 13.3% (2/15) at 3 months after vaccination]. The BNT vaccine group had higher median NAb titers than those in the AZ vaccine group at 1 month [median (IQR); 1:113 (1:11–1:160) vs. 1:10 (1:10–1:50), *p* = 0.002) and 3 months after vaccination [1:50 (1:10–1:131) vs. 1:10 (1:10–1:10), *p* = 0.003)] (Table 1).

All 30 serum samples collected prior to vaccination were sVNT-negative (<30% by GenScript and <25% by GenBody). Both GenScript and GenBody assays were positive in all samples one month post-vaccination, with median percentage inhibition results of 92.5% (IQR; 87.1–96.8%) and 96.8% (IQR; 91.2–98.6%), respectively. The percentage inhibition determined by sVNT at 3 months post-vaccination was significantly lower than that at one month post-vaccination in both AZAZ and BNT/BNT vaccine groups (*p* < 0.05) (Table 1 and Figure 1). Two participants (2/30) vaccinated with AZ showed discrepant results between PRNT and sVNT. One participant who was PRNT-positive with a 1:10 titer was sVNT-negative (17.0% in the GenScript assay and 21.3% in the GenBody assay). Another participant who was PRNT-positive (1:10) and GenScript-positive (47.0%) was GenBody-negative (18.2%) (Table A3). Overall, the titers measured by the two sVNTs showed excellent concordance with those obtained by PRNT (GenScript, 98.9%; k = 0.975; GenBody, 97.8%; k = 0.951) (Table A4). The BNT vaccine showed a higher inhibition rate (%) than that observed for the AZ vaccine at both one and three months post-vaccination.

### 3.2. Levels of SARS-CoV-2-Binding Antibodies 

In the four SARS-CoV-2 spike antibody serologic assays, antibody levels were compared according to the type of vaccine and blood collection time. Assay results for all 30 serum samples collected prior to vaccination were negative, indicating that antibody levels were below each manufacturer’s cut-off. For the 60 serum samples collected after vaccination, the Roche and LG S1 assay results were all positive. However, the LG RBD and Siemens assays showed negative results in 3.3% (2/60) and 1.7% (1/60) of the PRNT-positive samples, respectively. LG RBD showed negative results (ratios, 0.9 and 0.9) in two AZ-vaccinated participants who tested positive with a PRNT_50_ of 1:10 and sVNT of 25.0–47.2%. Siemens showed negative results (0.72 U/mL) in one AZ-vaccinated participant who was PRNT-positive (1:10) and sVNT-negative (17.0% in GenScript, 21.3% in GenBody) (Table A3). Overall, the results of SARS-CoV-2 binding antibody assays showed excellent concordance with those of PRNT (agreement 97.8–100.0%, k = 0.951–1.0) (Table A4). When comparing the AZ and BNT vaccine groups, binding antibody levels measured by Siemens, LG RBD, and LG S1 assays were higher in the BNT vaccine group than in the AZ vaccine group (*p* < 0.001). However, there was no significant difference in antibody levels between vaccine types using the Roche assay one month post-vaccination (*p* = 0.054) (Table 1).

### 3.3. Correlation of SARS-CoV-2 Antibody Assays with PRNT

The samples were subdivided into four groups according to the titer of PRNT50: <1:10 (*n* = 30), 1:10 (*n* = 30), 1:40 to 1:120 (*n* = 19), and >1:120 (*n* = 11). The results of the six serological assays were plotted according to the PRNT_50_ results and vaccine type (Figure 2). High PRNT_50_ titers were associated with high antibody titers. In samples with identical PRNT_50_ results, sVNT, CLIA, and ELISA showed higher antibody levels after BNT vaccination than after AZ vaccination. 

A Spearman rank correlation analysis of 90 PRNT-positive samples was performed to determine which serological assays were most highly correlated with PRNT. The sVNT results obtained by the GenScript and GenBody assays were strongly correlated with the PRNT_50_ results (*r* = 0.916 and *r* = 0.907, respectively) (Figure 3A). Results of the two sVNT assays (GenScript and GenBody) were also strongly correlated (*r* = 0.915). Results of CLIA and EIA targeting the RBD or S1 were also significantly correlated with those of PRNT50 (*r* = 0.860–0.901). There was a strong correlation between results of similar assays (*r* = 0.915 between sVNTs, *r* = 0.904 between CLIAs, and r = 0.910 between EIAs) (Figure 3B). Results of six serological assays showed significant correlations (*r* = 0.860–0.946). (Figure 4). 

## 4. Discussion

The presence of NAbs is an indicator of protective immunity against viral infection [16]. Among SARS-CoV-2 antibody assays measuring antibodies recognizing the S protein, only the NAb assay can reliably measure the actual protective immunity imparted by the antibody [17]. Notably, the SARS-CoV-2 RBD is immune-dominant and accounts for 90% of serum-neutralizing activity [18]. 

Many SARS-CoV-2 antibody assays have been developed, and the quality of serology tests vary widely. In addition, a variety of ELISAs have been developed as surrogate assays using recombinant RBD protein or ACE2 as the coating antigen. However, some commercial serological tests are limited by low sensitivity and specificity [19,20,21,22]. The low accuracy of serological tests is due to false positives caused by structurally altered antigens and epitopes, non-specific immunoglobulin binding, or protein-coated surfaces of ELISA plates [23,24,25].

While commercially available sVNTs are convenient and yield consistent results with those of standard PRNT, S1/RBD binding antibody assays using the principles of CLIA or EIA are high-throughput and more cost-effective for screening humoral immune responses in vaccinated populations. For CLIA, we analyzed converted results in BAU/mL, traceable to WHO international standards, which allow the accurate calibration of assays to arbitrary units, reducing variability among assay kits and leading to the harmonization of antibody levels [26].

According to Anton et al., surrogate ELISAs have two significant limitations. The synergism of antibodies targeting different epitopes cannot be explained, and detection is limited to antibodies that block the RBD/ACE2 interaction, missing other antibodies that neutralize via the non-RBD portion of the viral glycoprotein [27,28]. In fact, synergistic effects of antibodies targeting the RBD and S2 domain have been reported [29]. Surrogate assays are less sensitive than cell-based assays and may lead to more false-negative results [30].

To date, the gold standards for assessing the neutralization activity of drugs or antibodies are VNTs [31,32]. sVNTs are suitable for the rapid screening of large numbers of samples. Furthermore, as SARS-CoV-2 sVNTs are based on the antibody-mediated blockade of the ACE2-spike protein-protein interaction, they may contribute to the selection of convalescent plasma donors for the treatment of patients with COVID-19 [33].

GenScript sVNT, evaluated in this study, is an FDA-approved surrogate neutralization test that does not require virus or cell culture systems. Because it measures the ability to block RBD-ACE2 binding, this competitive ELISA is a surrogate for testing NAb activity [34]. This assay is now globally accepted as a more reliable test for NAbs than the SARS-CoV-2 binding antibody test. The GenBody sVNT uses recombinant hACE-2 protein as a coating antigen and fluorescent-conjugated recombinant S protein to detect neutralizing antibodies. Results can be obtained within 45 min using the GenBody sVNT, which can be performed as point-of-care testing. Our results demonstrated that GenScript and GenBody assay results are strongly correlated with each other and with the results of PRNT, consistent with previous results [11].

We used the ACE2-RBD binding inhibition rate and S1 or RBD binding antibody concentrations for evaluations. Two CLIAs and two EIAs showed excellent concordance with PRNT, and moderate correlations with PRNT_50_. In addition, these binding antibody assays showed strong correlations with commercial sVNTs in vaccinated participants, as observed in previous studies of patients with COVID-19 [35,36]. These results show that CLIA or EIA are alternative approaches to commercially available sVNTs, consistent with a previously proposed algorithm [37]. Therefore, the development of reliable alternative rapid tests with comparable results to those of sVNT or PRNT may provide a potential solution for COVID-19 antibody testing in small-scale laboratories.

The serum NAb titer after vaccination is an important indicator of vaccine efficacy. A 50% protective neutralization level is equivalent to an NAb titer between 1:10 and 1:30 in the serum, which is estimated to be approximately 54 IU/mL (95% CI 30–96 IU/mL) [38].

Our results provide the first quantitative comparison of S-specific or RBD-specific antibody levels with PRNT results using sequential samples obtained from vaccinated participants. Moreover, sVNT using ACE2-RBD inhibition and S1-or RBD-antibody assays based on CLIA or ELIA detected NAbs with good performance and generated results consistent with those of PRNT. Therefore, the six serological assays evaluated in this study represent potential alternatives for measuring NAbs against SARS-CoV-2 in vaccinated populations.

## Figures and Tables

**Figure 1 viruses-14-00946-f001:**
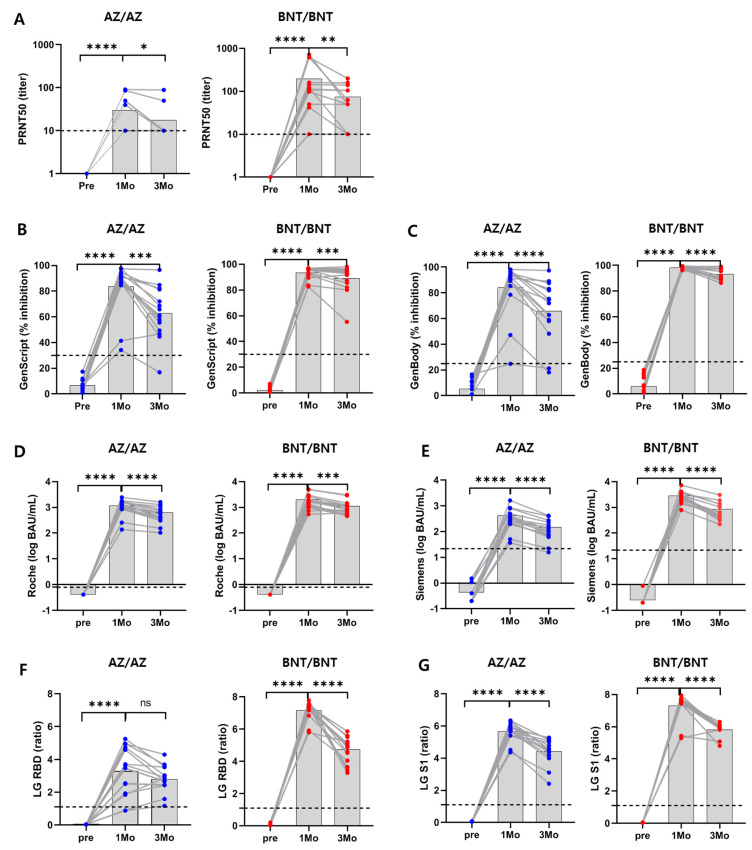
Anti-S1/RBD antibody and neutralizing antibody results for healthy donors after AZ/AZ or BNT/BNT vaccination. (**A**) Neutralizing antibody titers (PRNT); (**B**) ACE2-RBD inhibition by sVNT (GenScript); (**C**) ACE2-RBD inhibition (%) by sVNT (GenBody); (**D**) Anti-SARS-CoV-2 S RBD total Ig (log BAU/mL) (Roche); (**E**) SARS-CoV-2 S1 RBD IgG (log BAU/mL) (Siemens); (**F**) SARS-CoV-2 RBD IgG (ratio) (RBD) (LG RBD); (**G**) SARS-CoV-2 S1 IgG (ratio) (LG S1). Paired antibody courses at 1 month and 3 months post second dose are shown. Gray bars indicate median values. * *p* < 0.05; ** *p* < 0.01; *** *p* < 0.001; **** *p* < 0.0001 by Wilcoxon signed-rank test for paired samples.

**Figure 2 viruses-14-00946-f002:**
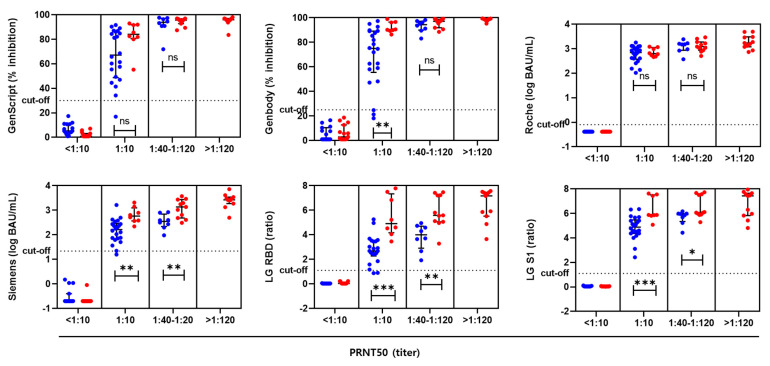
Levels of SARS-CoV-2 antibodies determined by six serological assays according to PRNT_50_ results. AZ vaccine group (blue dots) and BNT vaccine group (red dots) are indicated. ns, not significant; * *p* < 0.05; ** *p* < 0.01; *** *p* < 0.001 by Mann–Whitney tests.

**Figure 3 viruses-14-00946-f003:**
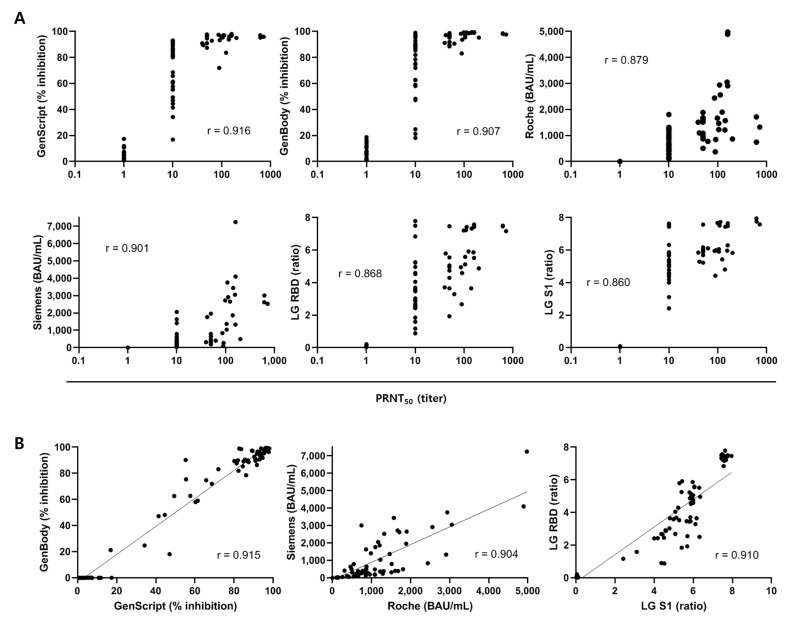
Correlations between antibody results. (**A**) Correlation between antibody levels by SARS-CoV-2 serologic assays (two sVNTs, two CLIAs, and two EIAs) and neutralizing antibody titers by PRNT. (**B**) Correlation between the antibody test results using similar assays (sVNT, CLIA, and EIA). Spearman correlation coefficients (*r*) are displayed.

**Figure 4 viruses-14-00946-f004:**
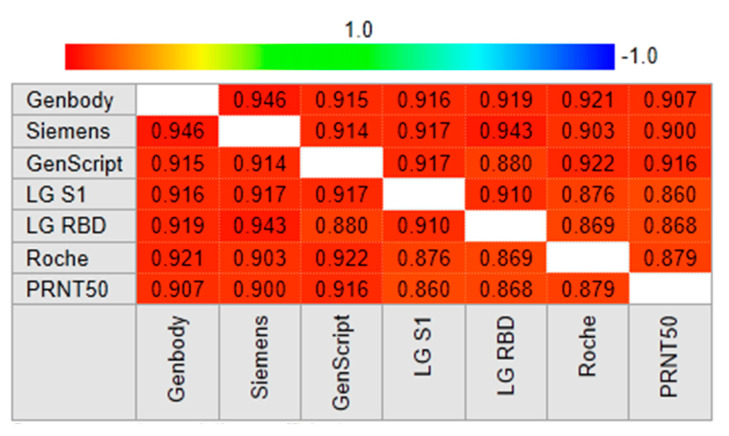
Spearman rank correlation coefficients for relationships between SAS-CoV-2 antibody assay results using 90 PRNT-positive samples from vaccinated participants.

**Table 1 viruses-14-00946-t001:** Comparison of SARS-CoV-2 antibody results by PRNT, two sVNTs, two CLIAs, and two EIAs in AZ and BNT vaccinated participants at 1 and 3 months after the second dose of the vaccine.

			AZ/AZ (*n* = 15)		BNT/BNT (*n* = 15)		
Sampling Time	SARS-CoV-2 Antibody Assays		Median	(IQR)	Median	(IQR)	*p*-value *
1 month after two-dose vaccination	PRNT	PRNT50 (titer)	1:10	(1:10– 1:50)	1:113	(1:44–1:160)	0.002
	sVNT	GenScript (% inhibition)	90.3	(86.4–92.9)	96.0	(92.3–97.0)	0.036
		GenBody (% inhibition)	91.2	(86.4–95.2)	98.6	(97.7–99.3)	<0.001
	CLIA	Roche (U/mL)	1236.0	(849.0–1473.5)	1622.0	(1086.5–2329.8)	0.054
		Roche (BAU/mL)	1270.6	(872.8–1514.8)	1667.4	(1116.9–2395.0)	0.054
		Siemens (U/mL)	14.6	(9.5–21.7)	121.9	(91.0–152.9)	<0.001
		Siemens (BAU/mL)	318.5	(208.1–472.4)	2658.2	(1984.8–3333.4)	<0.001
	EIA	LG RBD (ratio)	3.47	(2.08–4.70)	7.42	(7.18–7.49)	<0.001
		LG S1 (ratio)	5.78	(5.47–5.99)	7.57	(7.45–7.67)	<0.001
3 months after two–dose vaccination	PRNT	PRNT50 (titer)	1:10	(1:10–1:10)	1:50	(1:10–1:131.3)	0.003
	sVNT	GenScript (% inhibition)	61.4	(50.9–79.0)	93.8	(86.1–95.6)	<0.001
		GenBody (% inhibition)	71.9	(58.3–82.9)	91.8	(90.0–97.4)	<0.001
	CLIA	Roche (U/mL)	528.0	(369.3–803.3)	846.0	(601.5–1190.8)	0.021
		Roche (BAU/mL)	542.8	(379.6–825.7)	869.7	(618.3–1224.1)	0.021
		Siemens (U/mL)	4.9	(3.3–10.0)	23.7	(17.9–57.8)	<0.001
		Siemens (BAU/mL)	106.4	(72.4–218.4)	516.3	(390.8–1260.9)	<0.001
	EIA	LG RBD (ratio)	2.69	(2.44–3.41)	4.96	(4.17–5.45)	<0.001
		LG S1 (ratio)	4.58	(4.23–5.06)	5.88	(5.84–6.04)	<0.001

* Evaluated by the Mann–Whitney test. AZ, AstraZeneca ChAdOx1 vaccine; BNT, Pfizer BNT162b2 vaccine; N, number; IQR, interquartile range; PRNT, Plaque reduction neutralization test; sVNT, surrogate virus neutralization test; CLIA, chemiluminescent immunoassay; EIA, enzyme immunoassay; BAU, bound antibody.

## Data Availability

Not applicable.

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
