# Peer review of "Comparison of Six Serological Immunoassays for the Detection of SARS-CoV-2 Neutralizing Antibody Levels in the Vaccinated Population"

_viruses, 2022, doi:10.3390/v14050946_

Round 1
Reviewer 1 Report
The present paper by Lee et al evaluates the performance of six commercial immunoassays for neutralising antibodies to SARS-CoV-2. As gold standard, an in house plaque reduction test was used. As clinical material, cohorts fully vaccinated with vaccines from AstraZeneca and Pfizer-Biontech 1 and 3 months after booster dose were used. Over all, the experiments are well conceived and well executed and the results are clearly presented and intelligently discussed. The choice of well-defined clinical samples is excellent although it would have been interesting with the addition of comparative data from representative cases of immunity due to natural infection. Notably, for most assays and vaccine regimes, significant reductions are observed after 3 months with significantly higher effect for Pfizer-Biontech. Given the concern for waning immunity of vaccinated individuals, the topic has a broad interest for a general audience as well as clinical laboratories and in my opinion the paper should be published when the concerns below have been adequately addressed. My only major concern is point 1, all the other I consider minor. Since these experiments are already done, inclusion of these results in the paper should be doable in short time.
Specific points:
- As the authors’s point out, one limitation of the study is the number samples. Given the broad approach in terms of methodology this is fully acceptable for the positive samples but more worrisome for the negative samples. To verify specificity of diagnostic assays designed for a clinical setting a substantial number of truly negative samples are normally required (e.g. serum sampled from healthy blood donors taken prior the emergence of SARS-CoV-2 or equivalent). In this paper, the 30 samples taken prior vaccination were clearly tested in PRNT (line 140) and sVNTs (line 155) and serologic assays (line 177). Since some of samples from the vaccinated individuals were rather low (Fig 1), all samples taken prior vaccination should be presented in addition to the samples taken 1 and 3 months post vaccination to enable a proper comparison. Optionally, they could also be added to Fig A1.
- In results section (line 160) four patients with discrepant results are mentioned. In the text this is discussed in the context of Fig 1 but are identifiable in Fig 2. The discrepant results from Siemens and LG RBD (line 180) lacks reference to Fig 2. In general, a more generous reference to specific figure would facilitate for the reader. Since essentially all figures and tables show the same data in different ways, it is not always clear which figure/table is discussed.
- As mentioned above, it would have been interesting with an addition of comparative data from a representative cases of immunity due to natural infection. Although not absolutely essential, the usability of the study would have been broader.
- Whereas the paper is focused on quantitative comparisons (linearity, Spearman rank correlation and ROC-analyses), it would also have been interesting with a qualitative comparison (+/- exposure to antigen or neutralising immunity). For clinical applications, these is often a highly relevant question.
- In Fig 1, cut-off levels should have been indicated as are presented in Fig 2. For better readability of the low level signals, the scale in Fig 1B, left panel, should be adjusted. Although this will give a different scale when comparing against the right panel, this difference could be mentioned in other ways (insert of extra graph, interrupted scale with higher magnification at the lower end or an explaining sentence in the legend).
- Figure A1 and A2 is informative and could be considered for inclusion in the main paper.
- In Fig 2, the labelling of the square brackets (“**”, “ns” etc) could be positioned below rather than above the symbols.
- The rational for performing the ROC analysis could be a bit more extensive.
Reviewer 2 Report
The authors compare 2 CLIA, 2 ELISA and 2 sVNT to the PRNT, in order to evaluate serological assays in describing the antibody response to the vaccines by Pfizer-BioNtech and Oxford-AstraZeneca in the serum of 30 participants collected at 3 different time points (T0 before vaccination, T1 one month after 2nd dose and T2 three months after 2nd dose)
Strengths: Authors compare 6 commercially available kits that measure anti-SARS CoV 2 antibodies with different approaches. On the one hand there are binding assays (CLIA and ELISA) and on the other hand there are assays that measure neutralizing activity (sVNT). Comparative studies of different serological assays to gold standard tests are important for evaluating the most suitable tools that might help healthcare professionals in monitoring vaccination efficacy and defining vaccine strategies.
Weaknesses: The whole manuscript is confusing and unclear, some sentences are repeated and the results are not well-presented. There are many inaccuracies and assumptions that cause a somewhat superficial interpretation of the data.
For example, it is not clear whether the study is based on the assumption that all 6 tests under consideration measure the same biological marker, i.e. neutralizing antibodies. Binding assays return a measure of antibodies against S1 and RBD that is not associated with a neutralizing activity. It should be clearly stated in the whole manuscript that different assays measure different antibody-related parameters. In addition, if the authors want to provide a performance evaluation of the six serological assays, they should report performance analysis (precision/recall or sensitivity/specificity determination).
I also I suggest choosing the data to be represented in the figures more carefully and consistently with what is reported in the text.
English language requires extensive revision.
Overall, I think these data are important but they need to be more clearly presented.
Specific points:
- Line 39: please provide updated info and reference (https://covid19.who.int/)
- Lines 53-55: I am not sure that PRNT is the gold standard assay “to establish protective immunity”. I think protective immunity is something more complex that cannot be “established” only by monitoring neutralizing antibodies. If this is the case, please add a reference to this sentence. Otherwise, please use a more appropriate wording (for example “ to evaluate/determine neutralizing antibodies”). For reference please see doi: 10.1038/s41596-021-00536-y that you listed in bibliography.
- Table A1: info in Table A1 are not clearly reported
- Methods should be more detailed
- Lines 77-78: this sentence is not clear. When did sample collection was performed?
- Line 79; Please indicate the protocol of serum preparation and storage conditions of collected samples.
- Line 76 -77: Inclusion and exclusion criteria used to select eligible patients should be reported.
- Lines 120-121: Which neutralizing capacity? Binding assays do not measure neutralizing activity. Please, be clearer.
- For both sVNT and Binding antibody tests, authors should report if they performed replicated measurements.
- Table 1: sVNT do not measure titers but they return a percentage of ACE2-RBD binding inhibition. Also CLIA and ELISA methods do not measure neutralizing antibodies, but anti-RBD antibodies. Therefore, in my opinion the words “neutralizing antibody titers” in the Table legend are not appropriate. “Comparison of different anti-SARS-CoV-2 antibody assays…”
- Table 1: CLIA ROCHE (U/mL)Median: 1236.0 IQR: 849-473.5??? In addition, I agree that antibody titer should be expressed as a ratio. Nevertheless, in the case of antibodies against SARS-CoV2 measured by CLIAs the Coalition for Epidemic Preparedness Innovations (CEPI), the National Institute for Biological Standards and Control (NIBSC), and WHO have defined an International Standard (NIBSC code: 20/136), encouraging scientists to express anti-SARS-CoV2 Ig in BAU/ml or IU/ml. Please see also the paper by Kristiansen et al. (doi: 1016/S0140-6736(21)00527-4). Therefore, CLIAs results should be reported accordingly, and, also, a systematic discussion about the correlation between antibody titer expressed as ratio, antibody binding units and neutralizing activity should be provided.
- Lines 157-158: I suppose CI95 range is reported in the brackets. Isn’t it?
- Lines 158-159: In this sentence authors report sVNT results, but they refer to Figure 1, which instead represent antibody levels (expressed as U/ml and as ratio) obtained by CLIAs and ELISAs assays. Please correct this discrepancy. Additionally, In Figure 1 legend the words “neutralizing antibody response” should be removed.
- Please, refer paragraph 3.2 to a Table or to a Figure, otherwise it is difficult to follow the thread
- Figure 2: I find really interesting that in the 1:10 PRNT group the AZ vaccinated patients show variable neutralization activities ranging from 20% to >95% (as assessed by both Genescript and Genbody assays). Simlarly, both CLIA and ELISAs show that AZ patients display a more disparate response respect to BNT patients. This point should be better discussed and compared to similar or dissimilar data in literature. Please specify that CLIAs units are represented on a log axis.
- Line 205: what do the authors mean for “60 PRNT-positive 60 samples”?
- Please specify in all the Table and Figure Legends which statistical method was used to determine p value (if applicable)
- Lines 211: Unclear sentence, please clarify that you are presenting correlation between similar assays.
- Figure A1. For clarity, it would be better to rename with single letters each panel of Figure 2.
- Figure A2. Please modify PRT in PRNT
- Line 227, Lines 228-229 and line 294 are redundant
- Line 296: please modify 54 IU)/mL in 54 IU/mL
- The discussion section should be widely revised. In the first half authors substantially repeat results already presented with few references to similar works that have been already published. Overall, it appears that the conclusions are poorly supported by the results presented.
Round 2
Reviewer 1 Report
All my concerns have been adequately addressed and the paper is in my opinion now acceptable for publication. The only exception is my point 7, which was not in line with my intention but since this is a minor point of graphical matter, I leave it to the discretion of the copy editors of the Journal.